# CDK8-Novel Therapeutic Opportunities

**DOI:** 10.3390/ph12020092

**Published:** 2019-06-19

**Authors:** Ingeborg Menzl, Agnieszka Witalisz-Siepracka, Veronika Sexl

**Affiliations:** Institute of Pharmacology and Toxicology, University of Veterinary Medicine, 1210 Vienna, Austria; Ingeborg.menzl@vetmeduni.ac.at (I.M.); Agnieszka.Witalisz-Siepracka@vetmeduni.ac.at (A.W.-S.)

**Keywords:** CDK8, mediator complex, cancer

## Abstract

Improvements in cancer therapy frequently stem from the development of new small-molecule inhibitors, paralleled by the identification of biomarkers that can predict the treatment response. Recent evidence supports the idea that cyclin-dependent kinase 8 (CDK8) may represent a potential drug target for breast and prostate cancer, although no CDK8 inhibitors have entered the clinics. As the available inhibitors have been recently reviewed, we focus on the biological functions of CDK8 and provide an overview of the complexity of CDK8-dependent signaling throughout evolution and CDK8-dependent effects that may open novel treatment avenues.

## 1. Introduction

Cyclin-dependent kinases (CDKs) represent a protein family with a serine/threonine catalytic core that is controlled by the binding of regulatory subunits. As the activating partners are tightly controlled by cell cycle-dependent synthesis and degradation, they are known as cyclins. The first CDK, CDK1, was discovered in 1986 in *Schizosaccharomyces pombe* and *Saccharomyces cerevisiae* by the analysis of mutants with a defect in the cell cycle [1]. The cyclin-dependent kinase family rapidly grew, with newly identified members initially named on the basis of the amino acid sequence in their conserved domains. The confusion lasted until 1991, when a unifying nomenclature was established [2]. Although CDKs were initially described as regulators of cell-cycle progression, they have subsequently been shown to have diverse roles in biological processes such as metabolism, neuronal differentiation, hematopoiesis, angiogenesis, stem cell self-renewal, and spermatogenesis [3].

The budding yeast *S. cerevisiae* expresses six CDKs that may be divided into two general groups. Members of the first group bind multiple cyclins and guide cell-cycle progression. CDKs in the second group are activated by a single cyclin and regulate transcription [4]. Evolution has seen an increase in the number of CDKs, particularly in the number of cell cycle-related CDKs. In mammals, eight subfamilies containing 20 CDKs and up to 30 cyclins have been described [5]. CDK proteins range from 250 to more than 1500 amino acids in size. They have a typical two-lobe structure and a conserved catalytic core consisting of an adenosine tri-phosphate (ATP)-binding pocket, a PSTAIRE-like cyclin-binding domain, and an activating T-loop motif. In the absence of a cyclin, the catalytic cleft is closed by the T-loop and enzymatic activity is prevented. Interestingly, the so-called transcriptional CDKs show a higher degree of sequence conservation than cell cycle-related CDKs. Cyclins are more variable in sequence, although all of them are structurally defined by a cyclin-box domain. Cyclins vary in mass from 35 to 90 kDa (Figure 1) [6].

## 2. CDK8 as a Regulator of Transcription

### 2.1. CDK8 and the Mediator throughout Evolution

Together with cyclin C, MED12, and MED13, the CDK8 kinase forms the “CDK8 submodule” of the mediator complex. The remaining core mediator complex consists of 26 subunits that form a “head”, “middle”, and “tail” structure. The mediator complex builds a bridge for transcription factors, chromatin modifiers, promoters, and enhancers to RNA Polymerase II (RNA Pol II) and plays a central part during transcription. Depending on the stimulus and cell type, the CDK8 submodule reversibly interacts with the mediator complex to allow and modulate the function of transcription factors or chromatin modifiers.

Initial studies on yeast suggested that CDK8 functions largely as a transcriptional repressor. Phosphorylation of the RNA Pol II C-terminal domain (CTD) prior to assembly of the preinitiation complex (PIC) by the yeast CDK8 homolog suppressor of RNA polymerase B (Srb10) inhibits transcription in vitro [7]. Srb10 can directly antagonize transcriptional activators. In ideal growth conditions, Srb10 is active and regulates the turnover of the general control protein (Gcn4) and Ste12 by priming them via phosphorylation for ubiquitination and proteasomal degradation. The CDK8 homolog Srb10 also phosphorylates and causes nuclear export of the trans-activator Msn2, thereby inactivating it. Nutrient deprivation causes the degradation of Srb10, which in turn facilitates the stabilization of Ste12 and Gcn4, as well as the nuclear accumulation of Msn2, enabling the transcription of stress-induced genes [8,9]. The first hint that Srb10 has a positive effect on transcription came from the finding that it collaborates with Kin28 (the yeast CDK7 homologue) in RNA Pol II re-initiation processes [10]. Additional evidence for stimulatory effects on transcription stems from genome-wide chromatin immunoprecipitation (ChIP) studies, which revealed the presence of Srb10 at active and inactive genes in vivo [11,12]. Srb10 is now believed to coordinate yeast cell growth and to regulate adaptation to different environments by modulating the transition from growth into a stationary phase [13].

Cdk8 has a critical role during development of the metazoan model *Drosophila melanogaster* [14]. A recent study revealed opposing roles of TORC1 and Cdk8 in RNA processing. TORC1-dependent phosphorylation of Cdk8 induces its ubiquitination and degradation under optimal growth conditions. Starvation or inhibition of TORC1 induces the upregulation of Cdk8, which phosphorylates CPSF6 and thereby leads to a change in transcriptional programs with alternative RNA processing of transcripts to adapt autophagy, as well as nutrient and energy metabolism. The mechanism is conserved in mammals [15].

In mammals, the CDK8, MED12, and MED13 genes have been duplicated to create CDK19, MED12-like, and MED13-like (MED12L and MED13L) genes. CDK19 shares an 83% sequence homology with CDK8, while MED12 and MED13 share only a 59% and 53% sequence identity with their paralogues. As the paralogues bind mutually exclusively to the CDK8 submodule, different constellations have a great potential for functional specialization, which accounts for the complexity of the mediator. CDK8 is required for embryonic development: a complete knockout of CDK8 in mice results in embryonic lethality on day 2.5 to 3 due to a preimplantation defect [16]. However, conditional inducible deletion of CDK8 in adult mice does not induce any gross abnormalities, so CDK8 is considered dispensable for adult tissue homeostasis [17].

### 2.2. CDK8 in the Regulation of Gene Expression

CDK8 can both repress and activate gene transcription in mammals. As in yeast, CDK8 represses transcription in a kinase-dependent manner by phosphorylating cyclin H and thereby inactivating TFIIH [18]. A kinase-independent structural inhibition of transcription re-initiation upon binding of the CDK8 submodule to the mediator complex has also been postulated [19]. Mitogen-activated protein kinase (MAPK)-dependent phosphorylation and activation of the transcription factor CCAAT/enhancer-binding protein β (C/EBPβ) results in gene activation only when bound to a CDK8 submodule-free mediator complex [20]. The work has been extended by a study of retinoic acid (RA)-induced transcription of the retinoic acid receptor β 2 (RARβ2) gene. Activation depends on the assembly of a poised preinitiation complex on the RARβ2 promoter: the complex includes a mediator bound to the CDK8 submodule (CDK8-mediator). Upon stimulation by RA, the repressive CDK8 submodule is lost and transcription is initiated [21]. Binding of the CDK8 submodule to the “tail” domain of the core mediator via MED13 precludes RNA Pol II recruitment and this has been confirmed in subsequent studies [22]. In this situation, the CDK8 submodule acts as a switch to control interactions between the core mediator and RNA Pol II to regulate initiation and re-initiation events.

Besides initiating transcription, CDK8 modulates transcriptional elongation. CDK8-mediator functionally cooperates with the positive transcription elongation factor b (p-TEFb) to support transcription [23]. A similar mechanism accounts for the regulation of genes within the serum response network, where the CDK8-mediator regulates the assembly of the RNA Pol II elongation complex and enables the recruitment of p-TEFb and bromodomain containing 4 (BRD4) [24]. CDK8 appears to function similarly in modulating transcriptional elongation under hypoxic conditions. The hypoxia-induced factor 1 alpha (HIF1α) promotes the expression of genes involved in angiogenesis, glycolysis, metabolic adaptation, erythropoiesis, and cell survival [25]. To guarantee activation in a timely manner, target loci of HIF1α are loaded with transcriptionally-engaged, proximally-paused RNA Pol II. In hypoxia, the CDK8-mediator coordinates the recruitment of the super elongation complex (SEC) to promote transcriptional elongation of hypoxia-induced genes [26]. As such, CDK8 again functions as an adaptor to environmental conditions.

### 2.3. CDK8 as A Promoter of Transcription Factor Activity and Degradation

The CDK8 submodule can interact directly with transcription factors independently of the mediator complex to regulate signaling pathways including NOTCH-dependent signaling, transforming growth factor-β (TGF-β) and bone morphogenetic protein (BMP) receptor signaling, and signal transducer and activator of transcription (STAT) signaling. The NOTCH pathway is crucial for cell-cell communication, neuronal development, and T-cell differentiation [27,28,29]. Upon activation of the extracellular domain, proteolytic cleavage of the intracellular NOTCH domain produces the truncated intracellular NOTCH (ICN) peptide, which shuttles to the nucleus to activate transcription. CDK8 phosphorylates ICN to enhance its activity, but this modification also primes ICN for ubiquitination and subsequent degradation [30]. The TGF-β and BMP pathways propagate signaling through SMAD2 and SMAD3 downstream of TGF-β and through SMAD1, SMAD5, and SMAD8 downstream of BMP to control cell growth and fate [31]. Phosphorylation of the C termini of single mothers against decapentaplegic (SMAD) proteins results in an accumulation of SMAD proteins in the nucleus, where they promote transcription. The pathways are closely regulated by antagonists acting through mitogen-activated protein kinases (MAPKs), such as the epidermal growth factor (EGF) and fibroblastic growth factor (FGF), and by stress signals such as UV irradiation. The stimuli result in the linker-phosphorylation of SMAD proteins, muting the TGF-β and BMP response by retaining SMADs in the cytoplasm, where they are subject to proteasomal degradation [32,33]. In contrast, CDK8 and CDK9 phosphorylate the linker region before the assembly of SMAD proteins into transcriptional complexes in the nucleus. This enhances the transcriptional activity while simultaneously priming SMADs for eventual degradation [34].

The STAT proteins mediate cytokine responses. Upon stimulation, Janus kinases (JAKs) phosphorylate STATs, inducing their dimerization, nuclear translocation, and DNA binding, which facilitates the transcription of STAT target genes [35,36]. Interferon-γ (IFN-γ) signaling requires JAK-dependent STAT1 phosphorylation; fully-fledged STAT1-dependent transcription requires additional phosphorylation by CDK8 at the STAT1 serine 727 (STAT1^S727^). Besides STAT1, CDK8 also phosphorylates STAT3 and STAT5. Thereby, it is a key regulator of the IFN-γ-induced anti-viral response of macrophages [37]. As most of the described CDK8-dependent pathways are altered in cancer, it is not surprising that CDK8 has been proposed to contribute to tumor development. The following chapters describe CDK8’s oncogenic potential, which is summarized in Figure 2.

## 3. CDK8 in Solid Tumors

The first indication that CDK8 is a proto-oncogene came from work on colorectal cancer (CRC). The CDK8 gene was found to be amplified in 47% of 123 CRC patient samples [38] and subsequent cohort studies revealed a negative correlation between CDK8 gene expression and the survival of CRC patients [39]. An increased level of CDK8 was later found in advanced CRC stages III and IV, suggesting that CDK8 contributes to the progression of colorectal adenoma to carcinoma [40]. CDK8 has proto-oncogenic effects as it interacts with the wingless-related integration site (WNT) pathway to enhance the transcriptional activity of β-catenin. Besides directly promoting the transcription of β-catenin target genes via its phosphorylation, CDK8 also phosphorylates E2F1 on serine 375, thereby hindering the suppressive function of E2F1 on β-catenin. CDK8 knockdown in CRC cell lines reduces cell proliferation and decreases the number of cells in the G1 and S phase [38]. In contrast, CDK8 was shown to suppress tumor growth in an in vivo model of APC^Min^ colon cancer: the inducible deletion of CDK8 resulted in an increased tumor size and accelerated growth rates. The effect has been attributed to a reduction in histone H3K27 trimethylation and an upregulation of Polycomb group (PcG)-regulated genes involved in oncogenic signaling [17]. The apparent discrepancy in CDK8 function may reflect context-specific roles and emphasizes the diverse functions of the protein in transcriptional regulation.

CDK8 is also involved in the epigenetic changes in melanoma. An analysis of 36 melanoma patient samples revealed an inverse correlation between CDK8 and the histone variant macroH2A (mH2A) and a comparison of primary and metastatic melanoma samples showed a significant loss of mH2A upon invasiveness. MacroH2A is an epigenetic factor involved in the silencing of genes, including CDK8. Metastatic melanoma tissues harbor a predominantly methylated mH2A2 promoter region and elevated CDK8 levels. This association has given rise to speculation that CDK8 drives the progression of melanoma by enhancing cell growth and migration [41].

In breast cancer, the levels of histone variant mH2A1 are low and the F-box protein Skp2 is responsible for mH2A1 degradation. Together with Skp1 and cullin-1, Skp2 forms an E3 ligase complex (SCF complex) with an important role in regulating the cell cycle and maintaining genomic integrity [42]. A Skp2-mH2A1-CDK8 axis may be key to regulation of the G2/M transition, polyploidy, cell proliferation, and cell migration, and thus in the development of breast cancer. The overexpression of CDK8 and Skp2, accompanied by low levels of mH2A1, is associated with an adverse prognosis in breast cancer patients [43]. Analysis of the transcriptome data of 968 breast cancer patients has uncovered increased levels of CDK8/CDK19 paralleled by enhanced levels of cyclin C and MED13 compared to samples from patients with benign or hyperplastic breast cancers and from normal mammary tissues [44].

The function of CDK19 in breast cancer is not verified at present, but CDK8 acts downstream of the estrogen receptor as estrogen-induced transcription depends on CDK8 kinase activity. CDK8/CDK19 inhibitors suppress the growth of estrogen receptor-positive breast cancer cells and reduce the emergence of estrogen-independent cells [45]. Although the inhibition of CDK8/CDK19 fails to induce apoptosis, synergistic effects in combinatorial therapies might be beneficial for breast cancer patients. A combination of Senexin B (a CDK8/CDK19 inhibitor) with fulvestrant (a selective down-regulator of the estrogen receptor) shows an enhanced tumor-suppressive effect with no apparent toxicity in mouse models. The prevention of compensatory signaling mechanisms that depend on CDK8/CDK19 kinase activity represents a possible explanation for the synergy [24,26].

### CDK8 Triggers EMT and Invasiveness

Based on work in a pancreatic cancer model, CDK8 has been proposed to have a role in the epithelial-to-mesenchymal transition (EMT). Mutated K-RAS activates CDK8, which in turn triggers EMT partly via the WNT signaling pathway [46]. Studies on ovarian and pancreatic cancer cell lines have provided further evidence of a correlation between EMT-associated transcription factors and the expression of CDK8/CDK19. An analysis of the RNA-seq meta-data of 283 high-grade serious ovarian tumors has shown that high CDK8 levels correlate with a short relapse-free survival. Inhibition of CDK8/CDK19 by Senexin B or knockdown counteracts the signal rewiring of EMT-promoting factors and tumor cell invasion.

The invasiveness of pancreatic cancers is further induced by the BMP signaling pathway. An increased expression of BMP family members results in changes in cell morphology and enhanced cell motility that typically occur in EMT [47,48,49,50]. BMP-induced, SMAD1-driven upregulation of matrix metalloproteinase (MMP)-2 depends on CDK8 kinase activity. Indications of the mechanisms come from human pancreatic and ovarian cell lines and a murine breast cancer model, in which the inhibition of CDK8/CDK19 counteracts EMT by upregulating E-cadherin, downregulating Snail1 and Snail2, and reducing tumor cell invasion [51]. Angiogenesis is promoted via the CDK8-β-catenin-KLF2 axis and may increase the metastatic potential of pancreatic cancer. High levels of CDK8 promote the expression of vascular endothelial growth factor (VEGF), VEGFR2, MMP-9, c-myc, and cyclin D1 by upregulating β-catenin. β-catenin also lowers the expression of KLF2, facilitating angiogenesis [52].

Hepatic metastases are the leading cause of death in CRC [53]. CDK8 interacts with two signaling pathways to enable metastasis in a CRC model: it regulates MMP3 via the WNT/β-catenin pathway and induces miR-181b downstream of the TGF-β/SMAD pathway [54]. Micro RNA-181b inhibits the tissue inhibitor of metalloproteinases-3 (TIMP-3), which is a well-described suppressor of invasion, metastasis, and angiogenesis in CRCs [55]. Although CDK8/CDK19 kinase inhibition has no significant impact on the proliferation of colon cancer cells, it effectively suppresses hepatic metastasis, even when already established [54]. An analysis of data in The Cancer Genome Atlas (TCGA) shows that high CDK19 and CDK8 levels in prostate cancer are also associated with a high migratory potential. Knockdown or inhibition of CDK8/CDK19 kinase in prostate cancer cell lines has a relatively minor impact on cell viability, but significantly reduces migration and invasiveness [56].

The data suggest that CDK8/CDK19 inhibitors might have a therapeutic role. Although they have not been shown to induce apoptosis, they do reduce invasiveness and metastatic growth. The underlying mechanism is not yet fully understood, but the effects are consistently observed in a number of solid cancers.

## 4. CDK8 and Leukemia

CDK8-ChIPseq in the acute myeloid leukemia (AML) cell line of MOLM-14 cells (mixed lineage leukemia fusion MLL-AF9) has found CDK8 on super-enhancers (SE) paralleled by the binding of MED1 and BRD4. Gene ontology referred CDK8-occupied, SE-associated genes to hematopoiesis, cellular differentiation, and transcription. In MOLM-14 cells, CDK8 and CDK19 inhibit SE-associated genes, while BRD4 supports their transcription. Cortistatin A (CA), a natural product of a marine sponge with highly selective inhibitory effects on CDK8 and CDK19, reduces the growth of AML cell lines with a megakaryoblastic phenotype like MOLM-14 [57], possibly because of a blockade of STAT1^S727^ phosphorylation. Despite the fact that hyperactive STAT1 signaling is linked to the expansion of abnormal megakaryocytes in myeloproliferative neoplasm (MPN), inhibiting STAT1 tyrosine phosphorylation with the JAK1/2 inhibitor Ruxolitinib has no influence on megakaryocytic differentiation [58]. In contrast, JAK2-mutant MPN cells that display high STAT1^S727^ phosphorylation react to CDK8/CDK19 inhibition with growth arrest and differentiation. CA-induced differentiation is thus accompanied by the upregulation of key regulators of hematopoiesis and differentiation along the erythroid-megakaryocytic axis. Combining Ruxolitinib and CA may represent a new approach to treat JAK2-mutant MPN patients, as Ruxolitinib treatment alone is not curative [59].

A further inhibitor, SEL120-34A, has also been tested on AML cell lines. SEL120-34A reduces the phosphorylation of STAT1^S727^ and STAT5^S726^: Both of these sites are known targets of CDK8. Any anti-proliferative effect was restricted to AML cells with high levels of STAT1 and STAT5 phosphorylation and displaying hematopoietic progenitor markers. The results with CA in various AML cell lines were recapitulated with SEL129-34A, Senexin B, and CCT251545 (other ATP-competitive type I CDK8/CDK19 inhibitors). The cell line MOLM-14 represents an exception as the effects of CA were not shown by any other inhibitor [60]. The reasons for the discrepancy remain to be determined, but are most likely to relate to distinct selectivities for CDK8 and CDK19 or to off-target effects.

A further layer of complexity was recently added by the finding that the CDK8/cyclin C axis acts in a tumor-suppressive manner in T-cell acute lymphoblastic leukemia (T-ALL). Cyclin C-mediated activation of CDK8 interferes with the NOTCH pathway by enhancing ICN1 degradation. Cyclin C deficiency increases ICN1 oncogene levels and accelerates the progression of T-ALL [61]. This again highlights the context-specific roles of CDK8 which demand further investigations to dissect these differences and possibilities for novel treatment avenues.

## 5. CDK8 and the Immune System

In the absence of immunoregulatory cytokines, CDK8/CDK19 represses immune response genes. C/EBPβ is a transcriptional activator of acute phase responses such as innate and adaptive immunity, senescence, and receptor tyrosine kinase/Ras-mediated tumorigenesis [62]. Recruitment of PRMT5/WDR77 by CDK8/CDK19 triggers DNMT3A binding by the methylation of histone H4 arginine 3 (H4R3) and facilitates silencing by additional CpG island methylation around C/EBPβ binding sites [63]. CDK8/CDK19, C/EBPβ, and NF-κB are present at the promoters of inflammatory genes such as interleukin 8 (IL8), IL10, and the CC-chemokine ligand (CCL2) upon stimulation by Toll-like receptor 9 (TLR9).

CDK8/CDK19 also has positive effects on transcription, e.g., promoting TLR9-induced gene expression in myeloma-derived cells [64] and acting upstream of STAT1 to promote the transcription of anti-viral genes upon IFN-γ stimulation in macrophages [37]. CDK8 also acts upstream of STAT1 in natural killer (NK) cells, thereby modulating NK-cell cytotoxicity. Mice with a point mutation of serine 727 to alanine (Stat1^S727A^), the putative CDK8 phosphorylation site, are less susceptible to leukemia, other NK cell-surveilled tumors, and cancer metastasis. The effects have been observed in vivo and can be attributed to the enhanced cytotoxicity of Stat1^S727A^ NK cells [65]. CDK8 deletion in the NK cell compartment using Cdk8^fl/fl^ Ncr1Cre transgenic mice partially recapitulates the enhanced NK cell-mediated cytotoxicity and tumor surveillance by upregulating the cytolytic effector protein Perforin [66,67]. Unexpectedly, STAT1^S727^ phosphorylation is unaltered in CDK8-deficient NK cells, presumably because the function of CDK8 can be compensated for by CDK19. The exact mechanism of how CDK8 regulates NK cell activity is currently enigmatic and additional studies are required to enable us to predict whether inhibiting CDK8/CDK19 will be of therapeutic benefit.

## 6. CDK8 in Cancer Metabolism

The rapid proliferation of cancer cells requires enhanced levels of ATP, which are frequently provided by an increased glucose consumption and high rates of glycolysis. This phenomenon is described as the Warburg effect and may represent a therapeutic opportunity [68,69]. HIF1α is a master transcriptional regulator of glycolytic enzymes and acts synergistically with CDK8 to affect gene expression [26]. Follow-up studies have confirmed the key role of CDK8 in the expression of glycolytic genes and thus in the Warburg effect. Inhibiting CDK8/CDK19 in CRC cell lines lowers the levels of glucose transporters, glucose uptake, and glycolytic capacity and reduces cell proliferation and anchorage-independent growth. Transcriptome analysis showed that CDK8 kinase activity regulates several components of the glycolytic cascade [70]. Although inhibiting CDK8/CDK19 only has cytostatic effects on CRC cells, a combination of CDK8/CDK19 inhibitors with pharmacological inhibitors of glycolysis might represent a promising avenue for the treatment of highly glycolytic tumors.

Aberrant lipid and carbohydrate metabolism is another feature of cancer cells, although it is currently only poorly understood. The CDK8 submodule has a central role in the regulation of sterol regulatory element-binding protein (SREBP)-mediated de novo lipogenesis. Phosphorylation of SREBP-1c at T402 by CDK8 primes it for degradation to hinder the expression of lipogenic genes, de novo lipogenesis, and lipid accumulation in hepatocytes [71]. The mechanistic Target of Rapamycin (mTOR) signaling pathway is a key sensor of cellular energy/nutrient abundance and stress and altered functions are involved in the pathophysiology of diabetes, cancer, and aging [72]. In mouse models of obesity and aging, the activity of mTORC1 is inversely correlated with the level of the CDK8 submodule. This suggests that mTORC1 has a direct impact on the level of CDK8 protein, although the underlying mechanism remains unclear [73].

## 7. Potential Therapeutic Benefits of Targeting CDK8 in Cancer

Targeting CDK8 may be beneficial (I) by reducing the invasiveness and growth of solid cancers; (II) by reducing glucose uptake, leading to reduced levels of energy available to cancer cells; (III) by forcing cancer cells to differentiate as a result of the loss of self-renewal ability; and IV) by activating tumor surveillance by enhancing NK cell cytotoxicity (Figure 3). It may be difficult to generate CDK8-specific drugs that do not affect CDK19, but if the challenge can be overcome, it would allow CDK19 to compensate for the inhibition of CDK8 in the surrounding healthy tissue while eliminating cancer cells, which seem to depend specifically on CDK8.

The dynamic association of the CDK8 or CDK19 module with the mediator and the highly flexible composition of the mediator may either enhance or limit the potential of CDK8 as a drug target. Although cancer cells might depend exclusively on a mediator complex containing CDK8 to activate transcriptional programs such as the stemness program in AML and MPN [58,59,60], the fact that CDK8 is part of the mediator may enhance the side effects of treatment, although CDK8 inhibitors will presumably not disrupt steady-state transcription because untransformed cells have other mediator complexes to guarantee basic transcription. We believe that combinatorial treatment may allow a reduction of drug levels and lead to beneficial synergistic effects.

## Figures and Tables

**Figure 1 pharmaceuticals-12-00092-f001:**
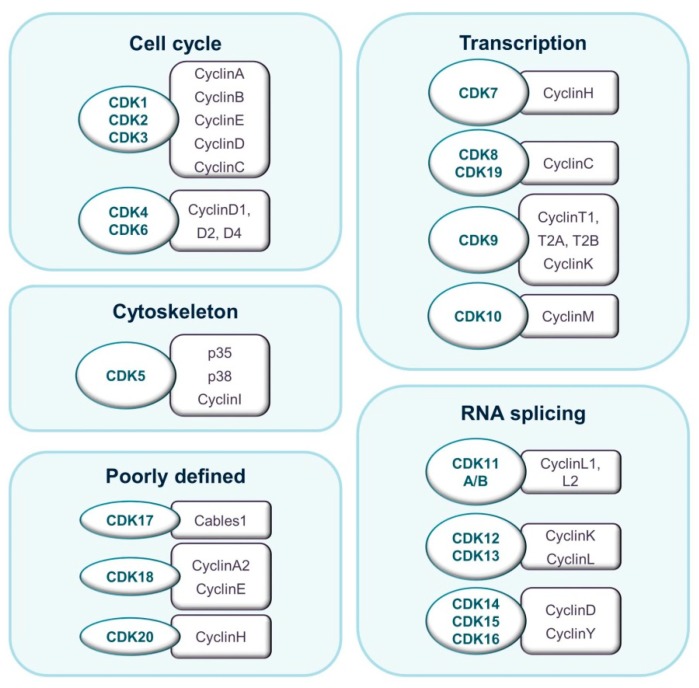
Classification of CDKs and their binding partners according to their prevalent described biological function.

**Figure 2 pharmaceuticals-12-00092-f002:**
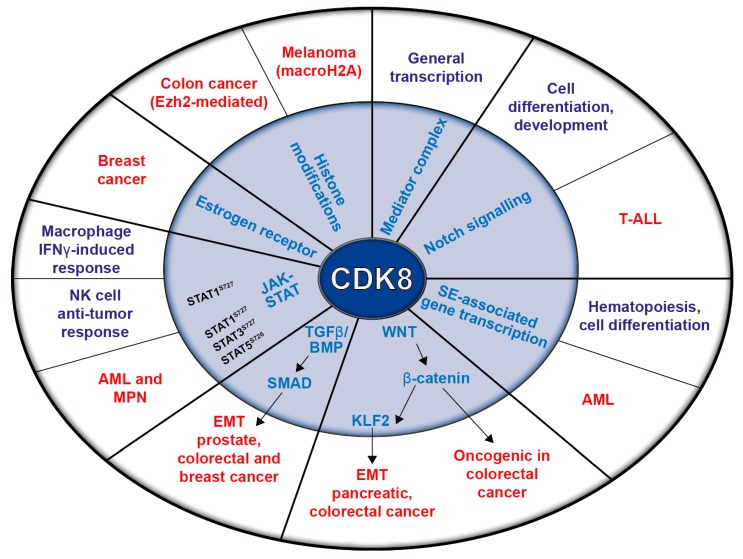
Schematic representation of CDK8’s functions in transcription and signaling pathways (inner circle) and their relation to physiological (violet) and pathological (red) conditions (outer circle).

**Figure 3 pharmaceuticals-12-00092-f003:**
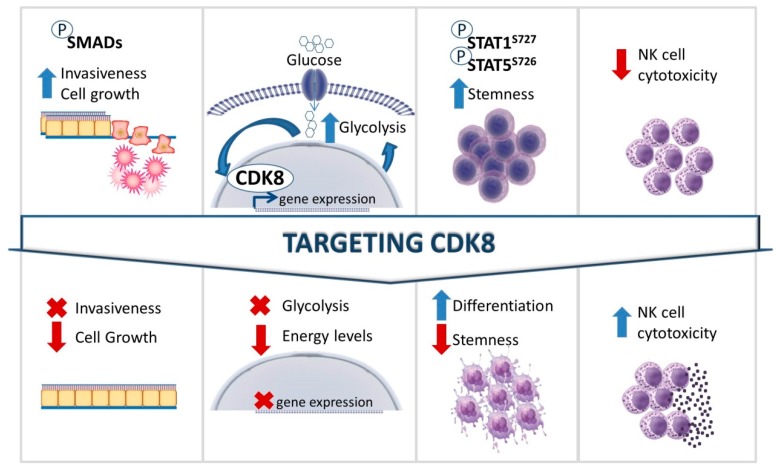
Potential therapeutic benefits of targeting CDK8 in cancer. Oncogenic functions of CDK8 are depicted in the upper panel. The lower panel shows the predicted outcomes upon targeting CDK8.

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
