# Peer review of "CDK8-Novel Therapeutic Opportunities"

_pharmaceuticals, 2019, doi:10.3390/ph12020092_

Round 1

Reviewer 1 Report

This is a nice review on CDK8. The authors have done a good job describing the different functions of CDK8.

One area of improvement could be to qualify some of the findings on CDK8. How likely are they to be true in the larger biological context. There are many findings that may not hope up the test of time. By structuring the review and focusing on the findings that the authors feel are the most important ones, would help the reader to understand the emphasis.

This is only a suggestion, which is not mandatory.

Author Response

We are delighted about the positive feedback of the reviewer. In the current version we put emphasis on the role of CDK8 in cancer and stratified the structure of the review accordingly. A native English speaker went through the manuscript. We are convinced that this makes the review easier for the readers to follow.

Reviewer 2 Report

Menzl et al. present a brilliant review on the roles of one of the major transcriptional kinases, Cdk8, in cancer. Following comprehensive overview, the authors scrutinize the involvement of Cdk8 and its cognate cyclin, cyclin C, in oncogenic signaling pathways pertaining to both solid and blood cancers. They correctly conclude that Cdk8/cyclin C may have both tumor-promoting and suppressing function depending on the biological context. The manuscript culminates with the depiction of Cdk8/cyclin C effects impinging on immune signaling and intermediary metabolism. The text is well written, easy to follow, and is accompanied with valuable explanatory figures. There are only few minor issues that can be easily taken care of listed below.

1) Could the authors please define the exact meaning of the term "CDK8-mediator" on its first

    occurence (line 94)?

2) Could the authors please arrange for a separate section entitled "CDK8 as promoter of  

    transcription factor activity and degradation" (line 111) with a correct section number and  

    the title highlighted using formatting consistent with other sections?

3) Could the authors please incorporate the definition of "type I CDK8/CDK19 inhibitor" (line 239)

    into the text?

4) Please assign a reference to "Li et al., 2014" on line 246 and replace this citation with the

    respective reference number in square brackets format.

5) Annotation of Figure 3 (lines 314-329) does not correspond to what is indicated in its legend.

    The legend to Figure 3 (lines 330,331) states "Kinase-independent and –dependent oncogenic

    functions of CDK8 are depicted in the upper panel.", however kinase-dependent functions are

    either missing or not properly labeled in the figure itself. Could the authors please explain

    or correct this concern?

6) There is a different formating style used between both the titles and text body parts of the

    Author Contributions, Funding, and Conflict of Interest sections (lines 333-338). Could the

    authors please make the formatting consistent among these sections?

There are also few corrections suggested below. Arrow => points to the correct version and the corresponding adjustment is indicated within a pair of angle brackets <>.

1) line 12: reviewed [1] we => reviewed [1], we <comma>

2) lines 21, 30, 33, 39: CYCLINS => cyclins <lower-case>

3) role => roles <plural>

4) lines 31, 36, 45, 87, 173, 243, 244, 245: CYCLIN => cyclin <lower-        case>

5) line 49: cell type the => cell type, the <comma>

6) line 68: melanogaster Cdk8 => melanogaster, Cdk8 <comma>

7) line 71: induces the upregulation => induces upregulation <no the>

8) line 73: autophagy, as well as => autophagy as well as <no comma>

9) line 78: CDK-submodule 8 different => CDK8 submodule, different  <CDK8 submodule,

                 comma>

10) line 80: development -complete => development - complete <space>

11) line 92: CDK8-module free => CDK8 submodule-free <submodule-free>

12) line 92: et al => et al. <period>

13) line 92: retinoic acid (RA) -induced => retinoic acid (RA)-induced
                   <(RA)-induced>

14) line 97: situation the => situation, the <comma>

15) line 100: [24] to actively support transcription => to actively support transcription [24] <citation

                     at the end of the sentence>

16) line 104: transcription CDK8 => transcription, CDK8 <comma>

17) line 110: (Galbraith et al. 2013) => [27]

18) line 113: NOTCH–dependent => NOTCH-dependent <hyphen>

19) lines 119, 126: TGF-ß => TGF-ß <ß instead of ß>

20) line 120: TGF- ß => TGF-ß <no space, ß instead of ß>

21) line 135: as key => as a key <a>

22) line 136: CDK8-involving => CDK8-dependent

23) line 142: patient samples of CRC => CRC patient samples

24) line 148: phosphorylation CDK8 => phosphorylation, CDK8 <comma>

25) line 150: S-phase => S phase <no hyphen>

26) line 154: (McCleland et al., 2015) => [18]

27) line 162: association CDK8 => association, CDK8

28) line 165: responsible E3 ligase => E3 ligase responsible

29) line 169: The overexpression => Overexpression <no the>

30) line 170: are associated => is associated <is>

31) line 171: cancer, where => cancer where <no comma>

32) line 176: receptor: as => receptor as <no colon>

33) line 178: estrogen independent => estrogen-independent <hyphen>

34) line 180: patient => patients <plural>

35) line 190: motility typically => motility, typically <comma>

36) line 197: EMT promoting => EMT-promoting <hyphen>

37) line 197: the cancer genome atlas => The Cancer Genome Atlas <capital     first letters>

38) line 220: CDK8-occupied SE-associated => CDK8-occupied, SE-associated     <comma>

39) line 227: (MPN) inhibiting => (MPN), inhibiting <comma>

40) line 234: inhibitor - SEL120-34A - was => inhibitor, SEL120-34A, was      <2x comma>

41) line 241: determined, distinct => determined; distinct <semicolon>

42) line 248: cytokines CDK8/CDK19 => cytokines, CDK8/CDK19 <comma>

43) line 264: tumor–surveillance => tumor surveillance <no dash>

44) line 285: CDK8-submodul => CDK8 submodule <no hyphen, submodule>

45) line 286: (SREBP) -mediated=> (SREBP)-mediated <no space>

46) line 291: disease mTORC1 => disease, mTORC1 <comma>

47) line 292: CDK8-submodule => CDK8 submodule <no hyphen>

48) line 292: levels, the => levels, however the <however>

49) line 299: tumor-surveillance => tumor surveillance <no hyphen>

50) line 302: CDK8 specific => CDK8-specific <hyphen>

51) line 308: like => such as

52) line 311: side–effects => side effects <no dash>

53) line 330: – dependent => -dependent <hyphen, no space>

54) line 333: I.M => I.M. <period>

55) line 333: A.WS => A.WS. <period>

56) line 333: V.S => V.S. <period>

Author Response

We are delighted about the positive feedback of the reviewer. We thank the reviewer for the detailed list of corrections and suggestions which have all been included in the revised manuscript.

Reviewer 3 Report

The manuscript by Menzl, Witalisz-Siepracka & Sexl entitled “CDK8 – novel opportunities” aims to give an overview on the complexity of CDK8-dependent signaling and effects, and its putative therapeutic targeting in cancer.

The review started nice but then very fast lost the focus; it reads like mixed styles from different writers making the visibility and getting into the topic difficult; for example: considering the abstract information and review's aims, it remains unclear why the topic of CDK8 in yeast and Drosophila are described in such detail. The manuscript will benefit from a more focused outline and unifying of the writing style.

CDK5 is a member of the family with a number of non-neuronal functions, which is not depicted on Figure.1. This requires correction.

I encourage the authors to provide several sentences with summary and conclusions as well as their own opinion of the reviewed data at the end of each chapter.

Author Response

We are grateful for the constructive feedback and have extensively worked on the structure and English of the manuscript. A native speaker has corrected the entire manuscript to unify the style. We agree that CDK5 has also a number of non-neuronal functions and revised the figure accordingly. Furthermore we revised the figure legend to point out that the depicted processes are just a rough categorization and show only a part of the functions.

Round 2

Reviewer 3 Report

Manuscript is improved